# Examining social isolation and loneliness in combination in relation to social support and psychological distress using Canadian Longitudinal Study of Aging (CLSA) data

**Verena H. Menec** [1] *, **Nancy E. Newall**[2], **Corey S. Mackenzie**[3], **Shahin Shooshtari**[1], **Scott Nowicki**[1]

1 Department of Community Health Sciences, University of Manitoba, Manitoba, Canada, 2 Department of Psychology, Brandon University, Manitoba, Canada, 3 Department of Psychology, University of Manitoba, Manitoba, Canada

* verena.menec@umanitoba.ca

## Abstract

### Background

Although a large body of research has focused on social isolation and loneliness, few studies have examined social isolation and loneliness together. The objectives of this study were to examine: 1) the relationship between four groups derived from combining social isolation and loneliness (socially isolated and lonely; only socially isolated; only lonely; neither socially isolated nor lonely) and the desire for more social participation, and social support; and 2) the relationship between the four groups and psychological distress.

### Methods

The study was based on the Comprehensive Cohort of the Canadian Longitudinal Study on Aging. Using CLSA baseline data (unweighted N = 30,079), ordinary and logistic regression analysis was used to examine the cross-sectional relationship between the four social isolation/loneliness groups and desire for more social participation and four types of social support (tangible, positive interaction, affection, and emotional support). Prospective logistic regression analysis was possible for psychological distress, which was derived from the Maintaining Contact Questionnaire administered about 18 months after the baseline questionnaire (unweighted N = 28,789).

### Results

Findings indicate that being socially isolated and lonely was associated with the most social support gaps; this group also had an increased likelihood of psychological distress, relative to those who were neither socially isolated nor lonely. Participants who were only socially isolated, and those only lonely also perceived some social support gaps. In addition, the only lonely group was more likely to be psychologically distressed than the only socially isolated group and the neither isolated nor lonely group.

**Data Availability Statement:** The data used in this study come from the Canadian Longitudinal Study on Aging (CLSA). All interested researchers can

access these data through CLSA, subject to review by the Data Access Committee, ethics approval, and signing of a data sharing agreement. Data are provided only once a data sharing agreement is in place between McMaster University (the custodian of the data) and the researchers' institution. For more information about data access, see https://www.clsa-elcv.ca/data-access. Applications are submitted to access@clsa-elcv.ca.

**Funding:** This research was made possible using the data/biospecimens collected by the Canadian Longitudinal Study on Aging (CLSA). Funding for the Canadian Longitudinal Study on Aging (CLSA) is provided by the Government of Canada through the Canadian Institutes of Health Research (CIHR) under grant reference: LSA 9447 and the Canada Foundation for Innovation. This research has been conducted using the CLSA Baseline Tracking Dataset 3.2, Baseline Comprehensive Dataset 3.1, under Application Number 170303. The CLSA is led by Drs. Parminder Raina, Christina Wolfson and Susan Kirkland. The analyses presented in this paper were funded by a CIHR grant (#373098) for V. Menec.

**Competing interests:** The authors have declared that no competing interests exist. The opinions expressed in this manuscript are the author's own and do not reflect the views of the Canadian Longitudinal Study on Aging.

## Conclusion

Examining the four social isolation/loneliness was useful, as it provided more nuanced risk profiles than would have been possible had we examined social isolation and loneliness separately. Findings may suggest avenues for interventions tailored to the unique needs of at-risk individuals.

## Introduction

Social isolation and loneliness have been studied extensively over the past four decades, with numerous studies showing that both have negative mental and physical health consequences [1–5]. Similarly, the factors that place people at risk of social isolation and loneliness have been studied extensively [6–8]. Policy makers are also increasingly recognizing that social isolation and loneliness are important public health concerns that need to be addressed. In Canada, the federal government has allocated funding for projects designed to reduce social isolation [9]. The United Kingdom has launched a loneliness initiative and implemented a Minister for Loneliness [10]. As part of this initiative, physicians will be able to offer "social prescribing" and direct patients to community workers who provide tailored support, for example.

Recently, we [11] argued that examining loneliness and social isolation in combination may provide important insights into people's social needs, and possible interventions that cannot be gained from examining the two concepts separately. The present study was designed to examine the association between four groups derived from combining social isolation and loneliness, and social factors and mental health in a sample of middle-aged and older Canadians who participated in the Canadian Longitudinal Study on Aging (CLSA). More specifically, given the CLSA data that were available at the time this study was conducted, we examined the cross-sectional relationship between social isolation and loneliness groups in relation to desire for more social participation, and perceived availability of social support, as well as their prospective relationship to psychological distress.

### Conceptualizing social isolation and loneliness

Consistent with previous literature, we conceptualized social isolation, loneliness and social support by drawing on distinctions between the structure (e.g., number of contacts; frequency of contact) and function (social support) of social networks [2,12–14], as well as the differentiation between objective assessments versus subjective perceptions of social relationships [14–16]. In line with this literature, social isolation can be defined in terms of people's social network structure, reflective of the objective state of a lack of social relationships [16]. In contrast, loneliness is typically defined as a subjective phenomenon that reflects the perception that emotionally intimate needs, or social needs are not being met [15–17]. As suggested by the discrepancy perspective of loneliness, loneliness arises when there is a mismatch between the quality and/or quantity of social relationships that people have versus what they want [17]. Hence, a person could be lonely despite having a relatively large social network, or be socially isolated and not feel lonely. That the two concept are distinct is supported by several studies that show that they are only weakly correlated with each other [18–21].

## The health-related outcomes of social isolation and loneliness

It is well established that social isolation and loneliness are health risks. For example, social isolation is associated with an increased risk of coronary heart disease and stroke [5], dementia [22], and mortality [2]. That social isolation is as much a risk factor for mortality as other well-known lifestyle risk factors, such as smoking, was noted over 30 years ago [12], a conclusion that was reiterated more recently on the basis of a meta-analysis of over a hundred studies [2]. Loneliness has also been shown to be associated with a wide range of physical and mental health outcomes, such as reduced cognitive function [23], depression [24–26], and mortality [3]. Moreover, both social isolation and loneliness have been shown to be related to negative health-related behaviors, such as reduced physical activity and smoking [24,27–29], as well as physiologic responses, including increased blood pressure and heightened inflammatory reactivity to stress [30–34]. Health behaviors and physiologic mechanisms are likely, in part, responsible for the negative health outcomes associated with social isolation and loneliness [31–34].

The relationship between social isolation/loneliness and social support has also been examined extensively, as it may be an important factor that ultimately leads to poor mental or physical health outcomes [12,34–37]. Social support refers to the types of assistance or help social network members provide, such as instrumental support with everyday tasks, or emotional support [12,13,36]. Both loneliness and social isolation are related to reduced social support availability [25,26]. For example, in a recent study, we [38] showed that a restricted social network structure, reflective of social isolation, was negatively related to several types of social support, including instrumental or tangible support (e.g., help with activities of daily living), emotional/informational support (e.g., having somebody to talk to or confide in), positive interactions (e.g., having somebody to have a good time with), and affectionate support (e.g., having somebody who gives love or affection). Lower perceived availability of certain types of social support was, in turn, related to increased depressive symptoms [39], a finding that is consistent with numerous studies that show that perceptions of having social support available is protective against depression [25,40].

## Social isolation and loneliness groups

Researchers have recommended that social isolation and loneliness should be examined together [3,21], and the two variables are sometimes included simultaneously in analyses [20,21,27,29,41–43]. However, few studies to date have examined the combined effect of social isolation and loneliness on health and health-related outcomes [11].

Victor and colleagues [44], based on research conducted in the 1950s and 1960s reported the prevalence of four social isolation/loneliness groups as 69% for people who are neither socially isolated nor lonely, 22% for the only socially isolated, 6% for the only lonely, and 3% for the socially isolated and lonely. However, no further information was provided regarding the four groups, such as how they differ in terms of their socio-demographic make-up or health outcomes.

Recently, Smith and Victor [45], using clustering analyses based on measures of social isolation, loneliness and living alone, found six groups of people. The largest group (47% of the sample) experienced no loneliness or social isolation. This group also had the least physical health problems and depressive symptoms. On the opposite side of the spectrum, a small proportion (4%) of the sample were both lonely and socially isolated and had a substantially increased likelihood of physical and mental health problems, relative to the majority group. In between these two extremes were four groups that experienced either social isolation or loneliness. These groups were also more likely to have physical and mental health problems

compared to the majority group. How these groups fared relative to each other on physical health or mental health was not examined, however.

In other relevant research, social isolation and loneliness were treated as continuous measures, and researchers examined how the interaction between the two measures relates to various social and health outcomes. Lee and Ko [46] examined the interaction between social isolation and loneliness in relation to people's social interactions. Among other measures, they assessed in detail the types of social interactions participants had in their daily lives, including whether they considered the persons they interacted with as 'weak' or 'strong' social ties. Results showed that greater loneliness, combined with a large social network was related to having fewer strong social network interactions. In other words, participants who were lonely, but not socially isolated, tended to have social interactions, but not with individuals they felt close to. From an intervention perspective, this suggests that these individuals may benefit from establishing new, meaningful relationships after, for example, loss of a spouse [46,47].

In other studies, researchers examined whether social isolation interacts with loneliness to impact mortality [18,20,48]. Beller and Wagner [18] showed a significant interaction, with the effect of social isolation becoming stronger at higher levels of loneliness, suggesting that being both socially isolated and lonely confers a much higher mortality risk than being only socially isolated or only lonely. However, no significant interaction between social isolation and loneliness on mortality was found in other studies [20,48].

## The present study

In the present study, we explored the relationship between social isolation, loneliness and social and mental health variables. Consistent with previous research [20,49–54], we treated social isolation and loneliness as dichotomous, rather than continuous variables, to create four social isolation/loneliness groups: 1) the 'socially isolated and lonely'; 2) the 'only socially isolated'; 3) the 'only lonely'; and, 4) the 'neither socially isolated nor lonely'. We took this approach, as it allows identifying individuals who might be most at risk of experiencing social support gaps and psychological distress and who may benefit from interventions that are tailored to their needs [55].

The objectives of this study were to examine:

1. the cross-sectional relationship between the four social isolation/loneliness groups and the desire for more social participation, and social support; and,

2. 2. the prospective relationship between the four social isolation/loneliness groups and psychological distress.

First, we focused on the desire for more social participation to gauge whether individuals in the different groups would actually want more social contact. Social isolation and loneliness interventions presume that people desire more social interaction; if they do not, however, then offers of help may be declined [11]. This issue is particularly relevant for the only socially isolated group, a group that has a small social network but does not feel lonely. Are individuals in this group perhaps choosing to have a small social network [47]? Given that they do not feel lonely, this group may lack the impetus to look for more social contact. Cacioppo and colleagues [31] argued that loneliness, like physical pain, serves an important role by functioning as an aversive signal that motivates people to reduce its negative consequences. Hence loneliness serves as a motivator to seek more social contact. If that is the case, then the only socially isolated may express little desire for more social participation, which may make them a difficult to reach group for interventions.

We hypothesized, therefore, that both the neither socially isolated nor lonely group and the only socially isolated group would be the least likely to want more social contact, but would not differ from each other. In contrast, given the double risk of social isolation and loneliness, the socially isolated and lonely group was expected to be the most likely to want more social interaction. The only lonely were also expected to want more social participation than the neither socially isolated not lonely and the only socially isolated, but less so than the socially isolated and lonely group.

Second, we examined the relationship between the four social isolation/loneliness groups and different types of social support (tangible, emotional, affective, and positive social interactions) [38,39,56], given the importance of social support in people's mental health [40]. Given that both social isolation and loneliness have been linked to less social support [25,38,39], we expected that the neither socially isolated not lonely group would report the most social support availability and the socially isolated and lonely group the least. As such, the socially isolated and lonely group, was expected to be the most at risk, similar to previous research that shows that social isolation and loneliness have a synergistic effect [18,45]. In considering the two middle groups (the only socially isolated and only lonely), we took into account that perceptions of social support, like loneliness, reflect individuals' appraisal of a situation [57]. We therefore expected loneliness to be more strongly associated with perceptions of social support than social isolation, with the only lonely group consequently reporting lower social support availability than the only socially isolated group.

Lastly, we examined whether the four social isolation/loneliness groups differ in terms of predicting psychological distress. As such, we build on an extensive literature that has examined social isolation and loneliness in relation to mental health [1,21,24,58]. Psychological distress includes symptoms of both depression and anxiety [59–61]. Given its association with psychiatric disorders and use of mental health services, psychological distress is frequently used as a measure of mental health in population studies [61,62]. Similar to social support, we expected the neither socially isolated nor lonely group to be least at risk of psychological distress and, conversely, the socially isolated and lonely group to be most at risk. The only socially isolated and only lonely were, again, expected to fall in between these two extremes, with the only lonely group expected to report more psychological distress than the only socially isolated group, given the strong association between loneliness and mental health [1,58].

The hypotheses can be summarized as follows:

1. Neither socially isolated nor lonely group = only socially isolated < only lonely < socially isolated and lonely for desire for more social participation.

2. Neither socially isolated nor lonely group > only socially isolated > only lonely > socially isolated and lonely for social support.

3. Neither socially isolated nor lonely < only socially isolated < only lonely < socially isolated and lonely group for psychological distress.

## Methods

### Data source

This study was based on data from the Canadian Longitudinal Study on Aging (CLSA) [63–65]. CLSA consists of a Comprehensive cohort and a Tracking cohort. Both cohorts were 45 to 85 years of age at baseline. Exclusion criteria for participation in CLSA were: not being able communicate in one of the two national languages (English or French); cognitive impairment at time of contact; resident of the three territories; full-time member of the Canadian Armed

Forces; resident in a long-term care institution at the time of recruitment; and living on Federal First Nations reserves or other First Nations settlements. All participants provided written consent for participation in the study. Further information about the CLSA and participant recruitment is published elsewhere [63–65].

In the present study, only the Comprehensive cohort was used (see below for rationale). The Comprehensive cohort consists of participants who were randomly selected within age/sex strata from among individuals residing within 25 to 50 kilometers of a CLSA data collection site in eleven locations across Canada (Victoria, Vancouver, Surrey, Calgary, Winnipeg, Ottawa, Hamilton, Montreal, Sherbrooke, Halifax, and St. John's). The participants were first interviewed in their own homes with computer-assisted interview instruments and subsequently came to a data collection site for additional computer-assisted interviews and comprehensive assessments, such as physical measures, and to provide biological samples.

Baseline data for the Comprehensive cohort were collected between May 2012 and May 2015. Approximately 18 months following the baseline interview, all participants (Comprehensive and Tracking) were re-interviewed via computer-assisted telephone interviews with the Maintaining Contact Questionnaire (MCQ). At the time the present study was conducted, only CLSA baseline data and data from the MCQ were available. The MCQ contains questions not asked at baseline. Although most of the measures in both the baseline and MCQ questionnaires are identical for Comprehensive and Tracking participants, some measures are included in one cohort only. This is the case for psychological distress, which was assessed only in the MCQ of the Comprehensive cohort. Because psychological distress was a key outcome measure in this study (and no other mental health measure was available in the MCQ), we restricted our analyses to the Comprehensive cohort. This allowed for an 18-month prospective analysis for psychological distress. As desire for more social participation and social support are assessed at baseline only, and are not included in the MCQ, analyses are cross-sectional for these outcome measures.

The present study received ethics approval from the University of Manitoba's Health Research Ethics Board. Data access was approved by the CLSA Data and Sample Access Committee.

## Study sample

The baseline data of the CLSA Comprehensive cohort contains 30,079 participants, representing over 3.7 million Canadians. Of these participants, 28,789 completed the MCQ. Given missing values on some measures, the actual sample size used differs somewhat across measures (see Analytic approach section for further information).

## Measures

**Social isolation/loneliness groups.** Four social isolation/loneliness groups were created by combining measures of social isolation and loneliness. Social isolation was defined based on a social isolation index that was derived based on individuals' contact with different social network groups. Including different social network groups is consistent with previous research that shows that a variety of social network members play an important role in people's lives, ranging from family members (spouses, children, siblings) to other relationships (friends, neighbors) [38,39], Similar to previous research [20,29,41,53,54], we allocated one point when each of the following conditions applied: 1) living alone and not married or in a common law relationship; 2) got together with friends or neighbours "within the past 6 months" or less frequently, or reported having no friends or neighbors; 3) got together with relatives/siblings "within the past 6 months" or less frequently, or reported having no relatives or siblings; 4) got

together with children "within the past 6 months" or less frequently, or had no children; 5) being retired and having participated in none, or only one of eight social activities at least once a month or more often (e.g., family or friendship based activities, church or religious activities, sports or physical activities, and educational and cultural activities) (for a more detailed description see 53 and 54). The resulting social isolation index ranged from 0–5, where higher scores reflect greater social isolation. The index was subsequently dichotomized. To allow a sufficient sample size in each of the four groups, individuals with scores 2 or more were classified as socially isolated and those with 0 or 1 as not socially isolated, similar to previous research [20,27,54].

The CLSA baseline questionnaire does not contain a loneliness scale. We therefore used a single-item loneliness question that is part of the CES-D10 depression scale [66]: [Over the past week] "How often did you feel lonely?" (1 = all of the time [5-7days]; 2 = occasionally [3–4 days]; 3 = some of the time [1–2 days]; 4 = rarely or never [less than 1 day]. Similar single-item measures are commonly used in the literature [1]. The item was dichotomized, with "all of the time" and "occasionally" responses classified as lonely and the remaining categories as not lonely.

The dichotomized social isolation index and loneliness measure were subsequently combined to create four groups: neither socially isolated nor lonely; only socially isolated; only lonely; and neither socially isolated nor lonely.

**Desire for more social participation.** In the baseline questionnaire, after responding to the eight questions related to social participation (see above for description), individuals were asked: "In the past 12 months, have you felt like you wanted to participate in more social, recreational, or group activities?" Yes/no.

**Social support.** Perceived social support was measured in the baseline questionnaire with the Medical Outcomes Study (MOS)–Social Support Survey [56]. This validated scale contains 19-items that focus on: tangible support (e.g., "someone to help you if you were confined to bed"); positive social interaction (e.g., "someone to get together with for relaxation"); affectionate support (e.g., "someone who hugs you"); and emotional/informational support (e.g., "someone you can count on to listen to you when you need to talk"). Responses are coded as: 1 = none of the time, 2 = a little of the time, 3 = some of the time, 4 = most of the time, 5 = all of the time. Mean scores were derived for each of the four subscales (range = 1–5).

**Psychological distress.** Psychological distress was measured in the MCQ questionnaire using the 10-item Kessler (K10) scale [59,60]. The scale is designed to assess non-specific psychological distress, with questions focusing on anxiety and depressive symptoms in the previous 30 days; e.g., "How often did you feel nervous?"; "How often did you feel hopeless?"; How often did you feel depressed?" [1 = none of the time; 2 = a little of the time; 3 = some of the time; 4 = most of the time; 5 = all of the time]. Summing across the 10 times creates a score ranging from 10 to 50. Consistent with previous research, we dichotomized the variable, such that participants with scores <20 were considered as having low distress and those with scores > = 20 as experiencing high distress [59,67,68].

**Covariates.** Several socio-demographic variables were included in the analyses, given their known relationship with social isolation and loneliness [6–8,53,54]. All variables were derived from the CLSA baseline questionnaire: age, sex, education, and household income. Age was categorized into four categories (ages 45–54, 55–64, 65–74, 75–85). Sex was coded as 0 = women and 1 = men. Education was dichotomized as: 0 = secondary school or less, and 1 = at least some post-secondary education. Household income was measured by asking participants to give the best estimate of the total household income received by all household members, from all sources, before taxes and deductions, in the past 12 months. The variable was categorized as 1 = less than $20,000; 2 = $20,000 or more, but less than $50,000; 3 = $50,000 or more; and "missing" for those who did not answer the question.

We also included several baseline health measures (chronic conditions, functional status, and depressive symptoms) in analyses, as physical and mental health is not only an outcome of social isolation and loneliness, but also a predictor [6–8]. For example, chronic conditions predict an increase in loneliness over time [69], and there is a reciprocal relationship between both functional limitations and loneliness, and depressive symptoms and loneliness [70].

The Older Americans Resources and Services (OARS) Multidimensional Functional Assessment Questionnaire [71] was used to assess functional status. The scale includes seven questions related to Activities of Daily Living (e.g., getting out of bed, dressing, and eating) and seven questions related to Instrumental Activities of Daily Living (e.g., using the telephone, shopping, and preparing meals). Participants were asked whether they can complete the task without help, with some help, or are completely unable to perform it. After categorizing individuals as having no functional impairment, mild impairment, moderate impairment, severe impairment, and total impairment, a dichotomized variable was created with: 0 = no functional impairment, and 1 = at least some functional impairment.

Chronic conditions were measured with a list of 33 conditions, such as osteoarthritis, respiratory conditions, and cardiac/cardiovascular conditions. For each condition, participants were asked if a doctor had diagnosed them with the condition. "Yes" responses were summed to create an overall index.

A baseline mental health variable was also included in multivariable analyses. Ideally, we would have included psychological distress as both a covariate and outcome variable. However, as that measure is not available in the baseline CLSA questionnaire, we used a single item from the CES-D10 scale instead: [Over the past week] "How often did you feel depressed?" (1 = all of the time [5-7days]; 2 = occasionally [3–4 days]; 3 = some of the time [1–2 days]; 4 = rarely or never [less than 1 day]. This single item was included, rather than the full CES-D10 scale, given that the loneliness item from the scale was used to derive our four social isolation/loneliness groups. We chose the depression item because it had the highest item to total CES-D10 scale correlation (r = .60). Turon and colleagues [72] recently concluded that a single item depression measure may be useful in ruling out the need for further psychological assessment or intervention.

**Analytic approach.**  Data were analyzed using survey weights provided by CLSA. As recommended by CLSA, we used trimmed weights in the descriptive analyses and analytic weights in inferential analyses [73]. All analyses were conducted using SAS version 9.4. First, we conducted bivariate analyses to compare the four social isolation/loneliness groups in relation to all the other variables. Because we were interested in all possible comparisons between the four groups, we used the LSMEANS option for regression analysis. The option calculated simultaneously all six possible pairwise comparisons between social isolation/loneliness groups. The generated parameter estimates are equivalent to those derived from individual regressions with different reference groups specified. The LSMEANS option for bivariate logistic regression was used to compare the four social isolation/loneliness groups in relation to dichotomous variables (sex, education, functional status, desire for more social participation, psychological distress). The option was used in bivariate ordinal regression involving ordinal variables (age group, household income), and in bivariate ordinary least squares regression for continuous measures (chronic conditions, depressive symptoms, social support). To reduce the likelihood of a Type I error, each set of multiple comparisons was evaluated for significance using a Bonferroni adjustment (p = .01 divided by 6 comparisons = .0017).

Second, the outcome measures (desire for more social participation, social support, and psychological distress) were analysed with the four social isolation/loneliness groups as key predictor using multivariable analyses, adjusted for covariates. The province of residence was also controlled for in these analyses, as recommended by CLSA when using survey weights

[73]. Again, the LSMEANS option was used to test all possible comparisons between the four social isolation/loneliness groups, using a Bonferroni adjustment (p = .01 divided by 6 comparisons = .0017). Missing values were deleted list-wise in the multivariable analyses. Missing values were minimal, ranging from <1% to 2.6%.

## Results

Descriptive statistics for the sample are shown in Table 1.

In this sample, 74.7% of participants were neither socially isolated nor lonely, 15.6% were only socially isolated, 6.7% were only lonely, and 3% were socially isolated and lonely (Table 2). Descriptive statistics (and associated bivariate analyses) for the four groups show some differences on covariates. For example, the neither socially nor lonely group was younger than the other three groups, although the latter did not differ significantly from each other.

Differences between the four social isolation/loneliness groups also emerged for our outcome measures in bivariate analyses (Table 2). For example, the neither socially isolated nor

**Table 1. Sample description.**

| Measures | Weighted N | Weighted % or Mean | Standard Error |
|---|---|---|---|
| **Age groups** | | | |
| Ages 45–54 | 1,563,066 | 42.0 | 0.38 |
| Ages 55–64 | 1,108,198 | 29.8 | 0.31 |
| Ages 65–74 | 638,899 | 17.2 | 0.23 |
| Ages 75–85 | 407,614 | 11.0 | 0.18 |
| **Sex** | | | |
| Female | 1,869,254 | 50.3 | 0.37 |
| Male | 1,848,523 | 49.7 | 0.37 |
| **Education** | | | |
| Less than postsecondary | 1,179,197 | 31.7 | 0.34 |
| Postsecondary | 2,538,392 | 68.3 | 0.34 |
| **Household income (yearly)** | | | |
| < $20,000 | 161,709 | 4.4 | 0.13 |
| $20,000 to < $50,000 | 655,621 | 17.6 | 0.25 |
| > = $50,000 | 2,694,978 | 72.5 | 0.30 |
| Missing response | 205,468 | 5.5 | 0.16 |
| **Functional status** | | | |
| No functional impairment | 3,413,505 | 92.3 | 0.18 |
| Mild, moderate, severe, total impairment | 286,007 | 7.7 | 0.18 |
| **Number of chronic diseases** | 3,744,848 | 2.94 | 0.02 |
| **Depressive symptom** | 3,734,419 | 1.29 | 0.00 |
| **Desire for more social participation** | | | |
| No | 1,886,290 | 50.5 | 0.37 |
| Yes | 1,849,218 | 49.5 | 0.37 |
| **Social support** | | | |
| Tangible support | 3,740,050 | 4.27 | 0.01 |
| Positive interactions | 3,740,399 | 4.26 | 0.01 |
| Affection | 3,740,263 | 4.49 | 0.01 |
| Emotional support | 3,740,602 | 4.23 | 0.01 |
| **Psychological distress** | | | |
| Low | 3,089,408 | 87.2 | 0.26 |
| High | 455,251 | 12.8 | 0.26 |

**Table 2. Social isolation/loneliness groups by covariates and outcomes measures: Weighted percentages (within each group) or weighted means.**

| | Total Sample | Neither isolated nor lonely | Only isolated | Only lonely | Isolated and lonely |
|---|---|---|---|---|---|
| | 100 | 74.7 | 15.6 | 6.7 | 3.0 |
| **Covariates** | | | | | |
| Ages 45–54 | 42.0 | 44.3[a] | 35.4[b] | 37.1[b] | 32.3[b] |
| Ages 55–64 | 29.8 | 29.4 | 30.4 | 31.9 | 31.9 |
| Ages 65–74 | 17.2 | 16.5 | 20.0 | 16.4 | 20.3 |
| Ages 75–85 | 11.0 | 9.8 | 14.1 | 14.6 | 15.5 |
| Women | 50.3 | 49.6[a] | 49.9[a] | 60.0[b] | 47.9[a] |
| Men | 49.7 | 50.4 | 50.1 | 40.0 | 52.1 |
| Low education | 31.7 | 30.8[a] | 31.2[a] | 39.9[b] | 40.0[b] |
| High education | 68.3 | 69.2 | 68.8 | 60.1 | 60.0 |
| < $20,000 | 4.4 | 2.1[a] | 10.2[b] | 7.6[b] | 21.7[c] |
| $20,000 < $50,000 | 17.6 | 14.8 | 25.4 | 25.3 | 31.9 |
| > = $50,000 | 72.5 | 78.0 | 58.0 | 60.4 | 38.5 |
| No response | 5.5 | 5.1 | 6.4 | 6.7 | 7.9 |
| No functional impairment | 92.3 | 93.7[a] | 89.6[b] | 86.4[c] | 82.2[c] |
| Mild/moderate/severe/total functional impairment | 7.7 | 6.3 | 10.4 | 13.6 | 17.8 |
| Number chronic conditions | 2.94 | 2.79[a] | 3.23[b] | 3.58[c] | 3.76[c] |
| Depressive symptom | 1.29 | 1.2[a] | 1.26[b] | 1.90[c] | 1.95[c] |
| **Outcome measures** | | | | | |
| No desire for more participation | 50.5 | 52.7[a] | 50.6[a] | 33.2[b] | 33.6[b] |
| Desire for more participation | 49.5 | 47.3 | 49.4 | 66.8 | 66.4 |
| Tangible support | 4.27 | 4.41[a] | 3.93[b] | 3.89[b] | 3.30[c] |
| Positive interaction | 4.26 | 4.39[a] | 4.04[b] | 3.76[c] | 3.37[d] |
| Affection | 4.49 | 4.65[a] | 4.16[b] | 4.09[b] | 3.42[c] |
| Emotional support | 4.23 | 4.33[a] | 4.06[b] | 3.78[c] | 3.49[d] |
| Low psychological distress | 87.2 | 89.9[a] | 86.9[b] | 66.6[c] | 62.7[c] |
| High psychological distress | 12.8 | 10.1 | 13.1 | 33.4 | 37.3 |

Percentages or means that do not share a superscript are significantly different from each other based on weighted, bivariate least square mean comparisons derived from regression analyses (logistic regressions for dichotomous outcomes, ordinary least squares regressions for continuous outcomes, and ordinal regressions for ordinal level outcomes) using a Bonferroni adjustment (p value of .01/6 = .0017).

lonely group reported more positive interactions and emotional support than the only socially isolated group, who in turn reported more positive interactions and emotional support than the only group who, in turn, reported more positive interactions and emotional support than the socially isolated and lonely group (see Fig 1). A slightly different pattern emerged for tangible support and affection, in that the only socially isolated and the only lonely groups did not differ from each other. (See S1 Fig for other measures).

We next conducted multivariable analyses for the three outcome measures. As the results shown in Table 3 indicate, including covariates in the analyses did not change the results for desire for more social participation. In both unadjusted and adjusted analyses, the neither socially isolated nor lonely group did not differ from the only social isolated group, but both groups were less likely to desire more social participation than the only lonely and the neither socially isolated nor lonely groups. The only lonely group and the neither socially isolated nor lonely group did not differ from each other.

Multivariable (adjusted) results also mirrored those from bivariate analyses for social support (Table 4) and psychological distress (Table 5). There was only one exception to this

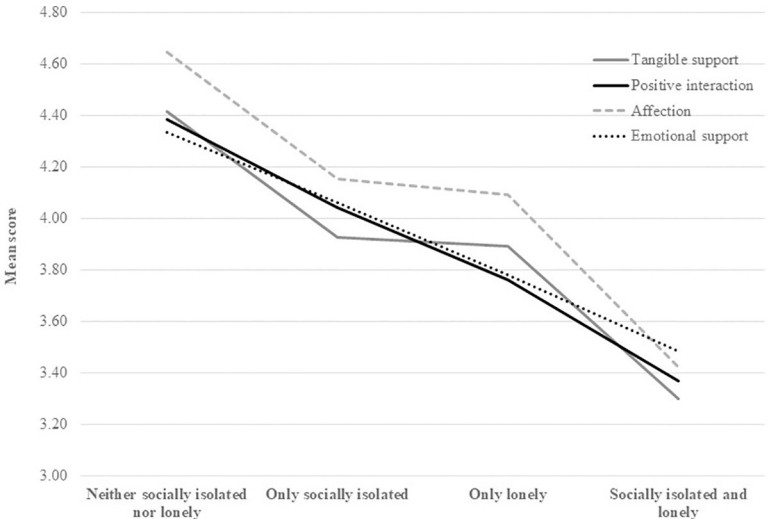

**Fig 1. Social isolation/loneliness groups by social support.**

pattern. For psychological distress, no difference emerged between the neither socially isolated nor lonely group and the only socially isolated group in adjusted analyses. A sensitivity analysis showed that the results for psychological distress were the same when the baseline depression measure was excluded from the analysis.

We further conducted sensitivity analyses to determine if results were similar for middle-aged (45–64 years old) versus older (65–85 years old) individuals, and women versus men, respectively (see S1 and S2 Tables). This was generally the case with one exception. Among individuals aged 65–85, no differences emerged between the only socially isolated group and the only lonely group for tangible support, positive interactions, and emotional support. However, the only socially isolated group reported receiving less affection than the only lonely group.

## Discussion

Although social isolation and loneliness have been examined alongside each other in previous studies [21,27,29,41,58], there is a paucity of research that has considered the two concepts in

**Table 3. Comparisons of social isolation/loneliness groups for desire for more social participation.**

| | Desire for more participation (vs. no desire) | |
| --- | --- | --- |
| **Social isolation/loneliness groups** | **Unadjusted[1]** | **Adjusted** |
| Neither isolated nor lonely vs. only isolated | -0.10 (0.03) | -0.10 (0.04) |
| Neither isolated nor lonely vs. only lonely | **-0.86 (0.05)** | **-0.78 (0.06)** |
| Neither isolated nor lonely vs. isolated and lonely | **-0.82 (0.07)** | **-0.71 (0.08)** |
| Only isolated vs. only lonely | **-0.76 (0.06)** | **-0.68 (0.06)** |
| Only isolated vs. isolated and lonely | **-0.72 (0.08)** | **-0.61 (0.08)** |
| Only lonely vs. isolated and lonely | 0.04 (0.09) | 0.07 (0.09) |

[1]Parameter estimates are shown and, in brackets, standard errors. The analysis is based on Baseline data, unweighted N = 29,784. Parameter estimates are derived from a weighted logistic regression and test differences in least squares means between groups. Adjusted analyses control for: age group, sex, education, household income, functional impairment, chronic conditions, depressive symptom, and province of residence at baseline. Statistical significance was assessed using a Bonferroni adjustment, p value of .01/6 = .0017. Significant results are bolded.

**Table 4. Comparisons of social isolation/loneliness groups for social support.**

| | Tangible support | | Positive interactions | | Affection | | Emotional support | |
|---|---|---|---|---|---|---|---|---|
| | Unadjusted[1] | Adjusted | Unadjusted | Adjusted | Unadjusted | Adjusted | Unadjusted | Adjusted |
| **Social isolation/loneliness groups** | | | | | | | | |
| Neither isolated nor lonely vs. only isolated | **0.47 (0.02)** | **0.39 (0.02)** | **0.34 (0.01)** | **0.26 (0.01)** | **0.48 (0.01)** | **0.40 (0.01)** | **0.27 (0.01)** | **0.20 (0.01)** |
| Neither isolated nor lonely vs. only lonely | **0.51 (0.02)** | **0.34 (0.02)** | **0.62 (0.02)** | **0.43 (0.02)** | **0.54 (0.02)** | **0.38 (0.02)** | **0.54 (0.02)** | **0.37 (0.02)** |
| Neither isolated nor lonely vs. isolated and lonely | **1.13 (0.04)** | **0.87 (0.04)** | **1.03 (0.03)** | **0.74 (0.03)** | **1.24 (0.04)** | **0.96 (0.04)** | **0.87 (0.03)** | **0.60 (0.03)** |
| Only isolated vs. only lonely | 0.04 (0.03) | -0.05 (0.03) | **0.28 (0.02)** | **0.17 (0.02** | 0.06 (0.03) | -0.02 (0.03) | **0.27 (0.02)** | **0.17 (0.02)** |
| Only isolated vs. isolated and lonely | **0.66 (0.04)** | **0.48 (0.04** | **0.69 (0.04)** | **0.48 (0.03)** | **0.76 (0.04)** | **0.56(0.04)** | **0.59 (0.03)** | **0.39 (0.03)** |
| Only lonely vs. isolated and lonely | **0.62 (0.04)** | **0.53 (0.04)** | **0.42 (0.04)** | **0.31 (0.04)** | **0.70 (0.05)** | **0.59 (0.04)** | **0.33 (0.04)** | **0.23 (0.04)** |

[1]Parameter estimates are shown and, in brackets, standard errors. The analyses are based on Baseline data, unweighted N = 29,644 for tangible support and positive interaction; unweighted N = 29,642 for affection; and unweighted N = 29,645 for emotional support. Parameter estimates are derived from weighted ordinary least squares regression and test differences in least squares means between groups. Adjusted analyses control for: age group, sex, education, household income, functional impairment, chronic conditions, depressive symptom, and province of residence at baseline. Statistical significance was assessed using a Bonferroni adjustment, p value of .01/6 = .0017. Significant results are bolded.

combination [18,20,45,46,48]. That differentiating between the four groups could be useful is based on the assumption that depending on whether people are socially isolated or lonely or both, they may have specific social support gaps and needs [11]. Understanding these gaps and needs could, therefore, provide valuable information to help tailor interventions. Conversely, it may help understand how to prevent social isolation and loneliness so that people remain neither socially isolated nor lonely.

The prevalence of the neither socially isolated nor lonely, only socially isolated, only lonely, and socially isolated and lonely groups was 74.7%, 15.6%, 6.7% and 3%, respectively, in the present study. These prevalence rates are quite similar to those reported by Victor et al. [44] for the four groups, which were 69%, 22%, 6%, and 3%, respectively. They are also in line with results by Smith and Victor [45]. In that study, 3.9% of the sample was both socially isolated and lonely and had a high likelihood of living alone. In contrast, less than half of the sample (47%) was neither socially isolated nor lonely. This lower prevalence compared to the findings in the present study could be due to a number of factors, such as different cut-offs to define

**Table 5. Comparison of social isolation/loneliness groups for psychological distress.**

| | High psychological distress (vs. low distress) | |
|---|---|---|
| **Social isolation/loneliness groups** | Unadjusted[1] | Adjusted |
| Neither isolated nor lonely vs. only isolated | **-0.30 (0.06)** | -0.11 (0.06) |
| Neither isolated nor lonely vs. only lonely | **-1.51 (0.06)** | **-0.66 (0.08)** |
| Neither isolated nor lonely vs. isolated and lonely | **-1.65 (0.08)** | **-0.65 (0.10)** |
| Only isolated vs. only lonely | **-1.21 (0.08)** | **-0.56 (0.09)** |
| Only isolated vs. isolated and lonely | **-1.35 (0.09)** | **-0.55 (0.11)** |
| Only lonely vs. isolated and lonely | -0.14 (0.10) | 0.01 (0.12) |

[1]Parameter estimates are shown and, in brackets, standard errors. Psychological distress is derived from the Maintaining Contact Questionnaire; unweighted N in the analyses = 28,042. Parameter estimates are derived from a weighted logistic regression and test differences in least squares means between groups. Adjusted analyses control for: age group, sex, education, household income, functional impairment, chronic conditions, depressive symptom, and province of residence at baseline. Statistical significance was assessed using a Bonferroni adjustment, p value of .01/6 = .0017. Significant results are bolded.

loneliness (Smith and Victor differentiated between moderately and highly lonely people), the measures included in their Latent Class Analysis (living alone was included as a separate variable), and the analytic approach per se (statistically derived groups versus an a priori classification used in the present study).

The four groups differed on socio-demographic characteristics in the present study, although the pattern of findings was not entirely consistent with previous research across all measures. The neither socially isolated nor lonely group was younger and had a higher household income than the other groups, similar to Smith and Victor's findings [45], but did not differ from the only socially isolated group in terms of sex and level of education. For the other groups, a few findings stand out. The only lonely group was composed of more women than the other groups, whereas the socially isolated and lonely group had the highest proportion of low-income individuals. In terms of baseline physical and mental characteristics, cross-sectional, bivariate analyses show that the neither socially isolated nor lonely group was less likely to be functionally impaired and had fewer chronic conditions and depressive symptoms than the socially isolated group, who in turn fared better on these measures than the only lonely group and the socially isolated and lonely group. The latter groups did not differ from each other. Overall, the findings suggest that the four groups had unique socio-demographic and health characteristics at baseline.

Turning to our outcome measures, our hypotheses were, in part, confirmed. Our findings indicate that being either socially isolated, lonely, or both, were all associated with some risk. The socially isolated and lonely group was clearly the most at risk group, as expected, consistent with previous research [18,45]. Although this group represents a small proportion of the population, they are an important target for intervention as they have the most social support gaps, and are likely to experience psychological distress. That individuals in this group were also likely to express a desire for more social participation is, on the one hand, positive as it suggests that they are motivated to have more social contact. On the other hand, the finding that a large proportion of individuals in this group are on low income and have health problems suggests that structural barriers to social engagement need to be addressed. For example, making affordable transportation options available could help them attend social programs. Programs that can be accessed from home may be also be appealing to this group. For example, technology interventions whereby individuals receive computer training or social interaction via videoconferencing have been found to reduce loneliness [74,75]. Low-tech, inexpensive approaches, including delivering programs via telephone may also be beneficial, particularly for home-bound individuals [76]. Home visits may also be useful for this group.

The only lonely were the second most at-risk group. This group also desired more social participation, reported some social support gaps and had a high likelihood of psychological distress. This is consistent with previous research that shows that loneliness has health-related impacts even when social isolation is controlled for [42,43]. Given that individuals in this group do not have their social needs met, despite having a relatively large social network, they may benefit from developing new friendships that are more meaningful to them, for example by exploring new activities that expose them to new social networks. For instance, interventions that provide activities, such as visual arts discussion or music participation show promise [74]. Barriers to social participation, such as functional limitations, may also need to be addressed by providing transportation. This group may also benefit from counselling to help improve existing relationships with others. For example, even though these individuals may have a spouse, the marital relationship may be strained, which can have major negative consequences for well-being [77]. This group may also benefit from interventions designed to address maladaptive expectation and social cognitions in relation to the people in their social network [55].

The only socially isolated group was expected to be somewhat at risk as well. Like the only lonely, they perceived some social support gaps relative to the neither socially isolated nor lonely group, but did not want more social participation. Although they were more likely to prospectively experience psychological distress than the neither socially isolated nor lonely group in unadjusted analyses, once we adjusted for baseline socio-demographic and health measures, this difference was no longer statistically significant. These findings suggest that pre-existing sociodemographic and health characteristics associated with social isolation, rather than social isolation per se, predicted psychological distress. This group, therefore, presents a somewhat mixed picture. That they report social support gaps places them at some risk should the need for help arise, a scenario that is not unlikely, given their health problems. That individuals in this group may not want more social contact could, at the same time, make them the most difficult to reach group from an intervention perspective, as they may decline opportunities for more social participation. The potential for these individuals to become lonely therefore exists. In this respect, it would be important in future research to examine people's group membership over time to determine what factors lead somebody to become both socially isolated and lonely.

The differences that emerged between the only socially isolated and only lonely groups warrant discussion at this point, specifically the differences for the four social support measures. We hypothesized that the only socially isolated would report relatively more availability of all four types of social support than the only lonely group. This hypothesis was only partially supported. As expected, the only socially isolated group reported more positive interactions and emotional support than the only lonely group. To the extent that loneliness reflects not having social needs met, it makes sense that the only lonely group rated their positive interactions and emotional support lower than the only socially isolated. The finding also underscores our conclusion above that the only lonely group could benefit from establishing new friendships. Previous research suggests that positive interaction is gained mainly from friends, rather than family [38].

However, the only socially isolated and only lonely groups did not differ from each other on tangible support. Tangible support focuses on the availability of somebody to help with daily activities (e.g., having somebody to prepare meals or help with daily chores). That the two groups did not differ for this type of support suggests that support with everyday tasks does not depend on the size of the social network [36]; presumably, as long as there is one person available, tangible support needs can, at least to some extent, be met. Another consideration is that we assessed the perceived availability of social support in the present study, not actual support received. An important issue for future research would be to examine what actual support people receive in situations of need. Do the only socially isolated have as much actual social support as they think they have? And what is the actual support available to the only lonely who, even though they have a relatively large social network, are not satisfied with it?

The two groups also did not differ in terms of affectionate support. Affectionate support focuses on having an intimate relationship with somebody; for example, having somebody who shows the person love and affection. Perhaps the two groups showed some affectionate support gaps (relative to the neither socially isolated nor lonely group) for different reasons; the only socially isolated because of their small social network, the only lonely because their social needs are not met, despite having a relatively larger social network. In this respect, it is also noteworthy that, although our sensitivity analyses showed similar results for women, men, and younger individuals, findings were somewhat different for older adults aged 65 to 85. Among these older individuals, the only socially isolated group reported less affectionate support than the only lonely group, but no differences emerged for tangible support, positive interactions, and emotional support. Older adults are at risk of losing their spouse or close friends. Such intimate relationships cannot easily be replaced; however, connecting individuals with social groups may satisfy at least some social needs. Overall, the complex findings suggest

that it would be important in future research to gain a more in-depth picture of relationship patterns and social support in different age groups, such as the quality of people's relationships, actual support received, and who provides support.

## Limitations

The present study was based on the Comprehensive Cohort of CLSA only, as psychological distress was measured only in this cohort. The Comprehensive cohort consists of randomly recruited participants residing within a certain radius of data collection sites in eleven Canadian cities. The results of this study may, therefore, not be generalizable to the Canadian population; for instance, our study includes only a small proportion of rural residents. The large sample size may also mean that although effects are statistically significant, they may not be clinically significant. Although CLSA contains measures of participants' social networks and social support, it does not include measures of the quality (positive, negative) of social relationships and actual support received. We were therefore not able to examine, for example, how the quality of relationships was related to social support gaps and whether, and what types of social support people received. Moreover, we classified participants in an a priori way into the four groups. It would be useful in future research to determine if the same groups emerge using clustering analysis. Using different cut-offs to identify socially isolated and lonely individuals could also change the results. Furthermore, the single item assessing the desire for more social participation does not allow examination of what types of social activities participants would have liked to engage in (e.g., more contact with family or friends, or more group activities). The baseline CLSA data also does not include the Kessler psychological distress scale. As such, we used a proxy mental health item (depressive symptoms) from the CES-D10 depression scale to control for baseline mental health in our prospective analyses. We were able to conduct a prospective analyses for psychological distress in the present study, as this measure was part of an 18-month CLSA follow-up questionnaire. However, this was not possible for social support and desire for more social participation, as follow-up data were not yet available at the time this study was conducted. It will be important in future research to examine social isolation/loneliness, social support, and mental and physical health trajectories over time. Examining whether social isolation and loneliness impacts people differently before and after retirement would also be useful, as retirement represents a major life transition.

## Conclusions

We draw several conclusions from our findings. First, examining social isolation and loneliness together was useful, as the findings suggest that there are some differences between the four groups. Second, the study suggests a number of areas for future study. For example, how does group membership change over time and what factors predict these changes? How does the quality (positive, negative) of social relationships intersect with social support and mental health? And what is the relationship between the four social isolation/loneliness groups and other outcome measures, such as physical health or life satisfaction? Third, examining the four social isolation/loneliness groups may suggest avenues for interventions tailored to unique needs. The present study represents a first step in this direction.

## Supporting information

**S1 Table. Adjusted analyses for middle-aged (age 45–64) versus older individuals (age 65–85).**
(DOCX)

**S2 Table. Adjusted analyses for women versus men.**
(DOCX)

**S1 Fig. Social isolation/loneliness groups by desire for more social participation and psychological distress.**
(JPG)

# Acknowledgments

This research was made possible using the data/biospecimens collected by the Canadian Longitudinal Study on Aging (CLSA). Funding for the Canadian Longitudinal Study on Aging (CLSA) is provided by the Government of Canada through the Canadian Institutes of Health Research (CIHR) under grant reference: LSA 9447 and the Canada Foundation for Innovation. This research has been conducted using the CLSA Baseline Tracking Dataset 3.2, Baseline Comprehensive Dataset 3.1, under Application Number 170303. The CLSA is led by Drs. Parminder Raina, Christina Wolfson and Susan Kirkland.

The analyses presented in this paper were funded by a CIHR grant for V. Menec (#373098).

# Author Contributions

**Conceptualization:** Verena H. Menec, Nancy E. Newall, Corey S. Mackenzie, Shahin Shooshtari.

**Formal analysis:** Scott Nowicki.

**Funding acquisition:** Verena H. Menec.

**Methodology:** Verena H. Menec, Nancy E. Newall, Corey S. Mackenzie, Shahin Shooshtari, Scott Nowicki.

**Supervision:** Verena H. Menec.

**Validation:** Scott Nowicki.

**Writing – original draft:** Verena H. Menec.

**Writing – review & editing:** Verena H. Menec, Nancy E. Newall, Corey S. Mackenzie, Shahin Shooshtari, Scott Nowicki.

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
