## [Decision Letter · Decision Letter 0]

2 Jan 2020

PONE-D-19-26452

Social isolation and loneliness groups: Associations with social support and psychological distress using Canadian Longitudinal Study of Aging (CLSA) data

PLOS ONE

Dear Dr. Menec,

Thank you for submitting your manuscript to PLOS ONE. After careful consideration, we feel that it has merit but does not fully meet PLOS ONE’s publication criteria as it currently stands. Therefore, we invite you to submit a revised version of the manuscript that addresses the points raised during the review process.

We would appreciate receiving your revised manuscript by Feb 16 2020 11:59PM. To enhance the reproducibility of your results, we recommend that if applicable you deposit your laboratory protocols in protocols.io, where a protocol can be assigned its own identifier (DOI) such that it can be cited independently in the future. For instructions see: http://journals.plos.org/plosone/s/submission-guidelines#loc-laboratory-protocols

We look forward to receiving your revised manuscript.

Kind regards,

Simone Reppermund, PhD

Academic Editor

PLOS ONE

Reviewers' comments:

Reviewer's Responses to Questions

**Comments to the Author**

1. Is the manuscript technically sound, and do the data support the conclusions?

Reviewer #1: Yes

Reviewer #2: Yes

Reviewer #3: Yes

Reviewer #4: Yes

2. Has the statistical analysis been performed appropriately and rigorously? 

Reviewer #1: Yes

Reviewer #2: N/A

Reviewer #3: Yes

Reviewer #4: Yes

3. Have the authors made all data underlying the findings in their manuscript fully available?

Reviewer #1: Yes

Reviewer #2: No

Reviewer #3: Yes

Reviewer #4: Yes

4. Is the manuscript presented in an intelligible fashion and written in standard English?

Reviewer #1: No

Reviewer #2: Yes

Reviewer #3: Yes

Reviewer #4: Yes

5. Review Comments to the Author

Reviewer #1: This is an interesting analysis that examines the meaning of the potential combinations of presence or absence of loneliness and social isolation in an adult population in Canada.

Specific comments

Title: You may consider deleting the word “groups” from the title, because the meaning may not clear to someone who is only viewing the title. It could mean an intervention group for this population. You mean subtypes rather than groups.

Abstract. The referral to “loners” or “lonely in a crowd” is not clear to the reader who has not yet read the introduction.

Page 4. Personally, I do not like the term “lonely in a crowd”. People who have a social network but are lonely are not “in a crowd”, but rather miss a certain intimacy within their social network which may not be that large, but does not qualify for social isolation.

Page 5 “the discrepancy theory of loneliness” sounds more like a definition than a theory.

Page 9, line 171 consider replacing ‘group’ with ‘groups’8

Page 34, in my opinion, the main limitations are not mentioned. Those pertain to

• dichotomizing continuous variables - . social isolation and loneliness are not dichotomous variables and using different cutpoint could change the results.

• Because of the large sample size, effect sizes could be very small despite significant results

• The sample mixes pre-retirement and post-retirement ages despite the differences between those age groups concerning processes concerning loneliness and isolation

• The percentage of lonely people in this sample (6.7%+3%=9.7% line 392 on page 20) is much lower than in most other reports of loneliness, especially in the older population.

Reviewer #2: This paper makes a significant contribution of characterizing the “isolated but not lonely” and the “lonely but not isolated” groups on sociodemographic variables and adjustment outcomes.

Introduction

Overall, the arguments can be streamlined by expressing the meaning of certain sentences and paragraphs more clearly.

1. Line 54: Minister “for” loneliness

2. The sentence from line 61 to 68 is unclear in meaning.

3. Line 71 to 11: it unclear whether the “on the one hand… on the other hand…” comparison meant to characterize the distinction between social isolation and loneliness only, or also between these two terms and social support. Hence the sentence need to be written to improve its meaning.

4. The paragraph from 152 to 161 can be written to express its meaning more clearly. The sentence from line 154 to 156 is particularly unclear.

Methods

5. The way in which participants have been split into four groups can be prone to type 1 error. Do the results hold if the analyses performed were continuous instead of categorical?

6. The authors mentioned that other similar studies have used cluster analysis. Have the authors tried cluster analysis on this dataset to see whether the four groups emerged?

7. Are the variables marital status and living arrangements (e.g. living alone vs with others) available in this dataset? These variables are quite important for studies on social isolation and loneliness.

Results

8. Could graphs be included for all outcome measures?

Discussion:

9. The take-aways, whether in terms of characterizations of or intervention for each group, needs to be made clearer. Currently, the findings read like lists of observations about each group. It will help the readers digest findings if the observations are organized in a more structured and coherent manner.

10. Technology-based interventions for the vulnerable group does not sound realistic or sustainable, given that this group has the largest proportion of “low education” and “low income”. Will regular home visits be a more suitable intervention?

11. Line 534 – 537: “Given that individuals in this group do not have their social needs met, despite having a relatively large social network, they may benefit from developing new friendships.” This sentence does not make sense. Why give people more of what is currently not working for them? It needs to be made clearer how the new friendships are different from existing ones.

12. Line 537: It is curious that the line “As for the vulnerable group…” appeared in the middle of a paragraph ostensibly about the lonely in a crowd group. Is the content in the rest of the paragraph that appears after this line about the vulnerable group or lonely in a crowd group?

13. What are the intervention implications of the results from age-based sensitivity analysis for older adults?

Reviewer #3: It is a great paper about the association of social isolation and loneliness with psychological distress, social support and civic participation. The topic is very interesting and there is a lack in the literature. However some aspects sohould improve:

1.-I think that the introduction should be shorter and focused on the topic (the association between social isolation and loneliness and their effect on civic participation, social support and health)

2.-It would be more clear to call the groups: 1)Neither lonely nor isolated 2)Only isolated 3)Only lonely 4)lonely and isolated.

3.-The authors should clarify the statistical analysis. I think that they used chi squared and anova tests for the comparisons in the descriptive analysis, adjusted and unadjusted ordered logistic regresssions models with the social support outcomes and, finally, adjusted and unadjusted logistic regressions models with participation and psychilogical distress as outcome. I also think that they used diferen sratifications of the samle to compare the effect between sepcific groups. But according to the methods section all this things are not clear.

4.-Authors should also clarify why the analysis wit psicological distress as outcome is prospective (like they say in the introduction)

5.-In my opnion, the main results are that loneliness but not social isolation showed a significant effect on psychological destress as well as that the effect of loneliness seem to be greater than those of social isolation in most of the outcomes.

6.-The authors should make further cites on rellevant literature such as:

Holwerda, T. J., Beekman, A. T. F., Deeg, D. J. H., Stek, M. L., van Tilburg, T. G., Visser, P. J., … Schoevers, R. A. (2012). Increased risk of mortality associated with social isolation in older men: only when feeling lonely? Results from the Amsterdam Study of the Elderly (AMSTEL). Psychological Medicine, 42(4), 843–853. https://doi.org/10.1017/S0033291711001772

Holwerda, T. J., Deeg, D. J. H., Beekman, A. T. F., van Tilburg, T. G., Stek, M. L., Jonker, C., & Schoevers, R. a. (2014). Feelings of loneliness, but not social isolation, predict dementia onset: results from the Amsterdam Study of the Elderly (AMSTEL). Journal of Neurology, Neurosurgery, and Psychiatry, 85(2), 135–142.

Reviewer #4: Review for PLoS One Journal

Social Isolation and Loneliness Groups: Associations with Social Support and Psychological Distress using Canadian Longitudinal Study of Aging (CLSA) Data

The purpose of this paper was to combine measures of social isolation and loneliness to create different groups and determine their association with social support and psychological distress. The four groups created were the majority group, the loner group, the lonely in a crowd group, and the vulnerable group (individuals in this group are dually socially isolated and lonely). In general, the majority group seemed to have the most positive and beneficial associations with social support and psychological distress.

I applaud the authors of this paper as it is innovative and will contribute to the social isolation and loneliness literature. The authors are correct in that many peer-review manuscripts have not examine social isolation and loneliness together, and this aspect of the manuscript alone greatly contributes to its innovation.

In my review of your manuscript, I have found some critiques which I believe will enhance the quality of the manuscript. The biggest issue in my opinion is regarding the methods. More detailed explanations are necessary for the type of analysis utilized, and also differentiating between the cross-sectional analyses and the prospective analyses.

The critiques of your manuscript organized by each section of your paper, from the abstract to the conclusion. I have included line numbers for where I find these issues.

ABSTRACT:

In the methods subsection of the abstract, it would be helpful to include the type of analyses done in the paper. This includes listing which analyses are prospective and which analyses are cross-sectional, and the type of regression modeling utilized (ordinary least squares, logistic, etc.)

INTRODUCTION:

Line 97: This study is actually over 30 years old now, so it should read over 30 years.

Lines 123-125: The authors raise a very important point here, in that most studies have not examined the interactive effects of social isolation and loneliness on health. That being said, the authors should also consider including studies by Cornwell and Waite (2009) and Coyle and Dugan (2012). These studies have simultaneously examined social isolation and loneliness (or Cornwell and Waite’s notion, social disconnectedness and perceived isolation) on health. Please include these studies when covering the literature.

Lines 162-167: Steptoe and colleagues (2013) also examined an interaction effect between social isolation and loneliness on mortality, and the interaction (I believe) was found to be non-significant.

Lines 260-263: There needs to be substantially more information on the amount of missing here. In particular, which measure had the most missing and what measure had the least missing? Were the missing enough to substantially affect the analysis? In general, if the missing are greater than 10%, this can represent a significant problem in the analysis.

Also, how were missing data handled? (I assume through listwise deletion?) If this is the case, then it should be stated.

Also, a final sample size should be stated for each of the analyses.

METHODS

In lines 354-363, the authors discuss having a baseline mental health variable to control for the effects of mental health at baseline. I agree with the authors and think, overall, it would be better to include psychiatric distress at baseline and as an outcome. That being said, I believe it would be helpful to include (in supplementary analyses or aside from the main analyses) another set of analyses without the baseline mental health variable. It would be helpful to do this to determine the consistency of the results and to determine if any of the relationships change.

In lines 380-385: There needs to be a stronger description of the analytic methods used in these analyses. If these are prospective analyses, then that should be stated. For example, if this is a lagged variable analysis or hierarchical modeling analysis, then this should be stated as well.

RESULTS

No changes necessary.

DISCUSSION

Overall, the discussion is written very well.

From lines 537 to 543 is somewhat of a weird transition. A few lines above, it starts with the most vulnerable group, and then it transitions to the lonely in a crowd group. From 537-543, it then goes back to the most vulnerable group. I would suggest discussing one group at a time instead of going back and forth between groups to increase clarity.

6. PLOS authors have the option to publish the peer review history of their article (what does this mean?). If published, this will include your full peer review and any attached files.

Reviewer #1: No

Reviewer #2: No

Reviewer #3: No

Reviewer #4: No

---

## [Author Response · Author response to Decision Letter 0]

28 Jan 2020

PONE-D-19-26452

Social isolation and loneliness groups: Associations with social support and psychological distress using Canadian Longitudinal Study of Aging (CLSA) data

PLOS ONE

Dear Dr. Reppermund;

We have revised our manuscript in line with the two Reviewers’ helpful comments. Below please find responses to the Reviewers’ comments. Please note that the page numbers indicated in our response refer to the unmarked manuscript without tracked changes. 

Thank you for re-considering our paper.

Reviewers' comments:

Reviewer's Responses to Questions

Comments to the Author

1. Is the manuscript technically sound, and do the data support the conclusions?

Reviewer #1: Yes

Reviewer #2: Yes

Reviewer #3: Yes

Reviewer #4: Yes

2. Has the statistical analysis been performed appropriately and rigorously? 

Reviewer #1: Yes

Reviewer #2: N/A

Reviewer #3: Yes

Reviewer #4: Yes

3. Have the authors made all data underlying the findings in their manuscript fully available?

Reviewer #1: Yes

Reviewer #2: No

CLSA data are available via application to the Data Access Committee (see https://www.clsa-elcv.ca/data-access). Data are made available to researchers through a data sharing agreement for the purposes of a specific project and cannot be shared with others. 

Reviewer #3: Yes

Reviewer #4: Yes

4. Is the manuscript presented in an intelligible fashion and written in standard English?

Reviewer #1: No

Reviewer #2: Yes

Reviewer #3: Yes

Reviewer #4: Yes

5. Review Comments to the Author

Reviewer #1: This is an interesting analysis that examines the meaning of the potential combinations of presence or absence of loneliness and social isolation in an adult population in Canada.

Specific comments

Title: You may consider deleting the word “groups” from the title, because the meaning may not clear to someone who is only viewing the title. It could mean an intervention group for this population. You mean subtypes rather than groups.

Good point. We have changed the title of the paper to: “Examining social isolation and loneliness together in relation to social support and psychological distress using Canadian Longitudinal Study of Aging (CLSA) data”

Abstract. The referral to “loners” or “lonely in a crowd” is not clear to the reader who has not yet read the introduction.

We are now referring to the four groups more descriptively as ‘socially isolated and lonely’; ‘only socially isolated’; ‘only lonely’; and, ‘neither socially isolated nor lonely’ throughout the paper. 

Page 4. Personally, I do not like the term “lonely in a crowd”. People who have a social network but are lonely are not “in a crowd”, but rather miss a certain intimacy within their social network which may not be that large, but does not qualify for social isolation.

As noted above, we are no longer using any labels for the four groups. 

Page 5 “the discrepancy theory of loneliness” sounds more like a definition than a theory.

We have changed this to ‘perspective’, a term that is commonly used in the literature when describing the discrepancy perspective of loneliness (p. 5). 

Page 9, line 171 consider replacing ‘group’ with ‘groups’8

This typo has been corrected. 

Page 34, in my opinion, the main limitations are not mentioned. Those pertain to

• dichotomizing continuous variables - . social isolation and loneliness are not dichotomous variables and using different cutpoint could change the results.

• Because of the large sample size, effect sizes could be very small despite significant results

• The sample mixes pre-retirement and post-retirement ages despite the differences between those age groups concerning processes concerning loneliness and isolation

We have added these points in the limitations section (pp. 34/35).

• The percentage of lonely people in this sample (6.7%+3%=9.7% line 392 on page 20) is much lower than in most other reports of loneliness, especially in the older population. 

The prevalence of loneliness varies widely in the literature, depending on the sample, measure, and cut-off used. Our cut-off is quite stringent; i.e. participants who indicated that they are 3 or more days per week lonely were identified as being lonely. Moreover, our sample included both middle-aged and older adults. Our prevalence estimate of 9.7% lonely is not inconsistent with previous research; e.g., Beutel et al. (2017) found a loneliness prevalence of 10.5% in a sample of 35-74 year olds; Wenger & Burholt (2004), in a sample of older adults, found a prevalence of 9% who were very lonely; and Dykstra (2009) showed a wide range in the prevalence of loneliness across countries: <5% in Denmark; 5-9% in Finland, Germany, Netherlands, and UK; 10-14% in Belgium, France, Ireland, Luxembourg, and Spain; and >19% in Greece and Portugal. 

Reviewer #2: This paper makes a significant contribution of characterizing the “isolated but not lonely” and the “lonely but not isolated” groups on sociodemographic variables and adjustment outcomes.

Introduction

Overall, the arguments can be streamlined by expressing the meaning of certain sentences and paragraphs more clearly.

1. Line 54: Minister “for” loneliness

This mistake has been corrected.

2. The sentence from line 61 to 68 is unclear in meaning.

We have revised this section to make the meaning clearer. 

3. Line 71 to 11: it unclear whether the “on the one hand… on the other hand…” comparison meant to characterize the distinction between social isolation and loneliness only, or also between these two terms and social support. Hence the sentence need to be written to improve its meaning.

We have revised this sentence.

4. The paragraph from 152 to 161 can be written to express its meaning more clearly. The sentence from line 154 to 156 is particularly unclear.

We have revised this section.

Methods

5. The way in which participants have been split into four groups can be prone to type 1 error. Do the results hold if the analyses performed were continuous instead of categorical?

We did not conduct analyses with social isolation and loneliness treated as continuous measures for two main reasons: 1) As noted in the paper (p. 8), social isolation and loneliness are often used as dichotomous variables. This approach, as opposed to keeping then continuous variables, is particularly useful as it allows identifying at-risk individuals who could, potentially benefit from interventions. 2) Creating four groups by combining social isolation and loneliness follows the same logic; i.e. it allows identifying specific groups of individuals who may be at risk. We have now revised the introduction to make this rationale clearer (pp. 8/9). 

In terms of type 1 errors, we certainly agree with the Reviewer. This is why we used a conservative Bonferroni adjustment to test for significant differences between the four groups for each of the outcome measures, with each test evaluated at an alpha of .01/6=.0017. That is, our family-wise error rate is controlled at alpha of 01. Given the conservative nature of the Bonferroni adjustment, it is possible that we therefore committed type 2 errors. 

6. The authors mentioned that other similar studies have used cluster analysis. Have the authors tried cluster analysis on this dataset to see whether the four groups emerged?

We have at this point not conducted cluster analysis to determine whether the four groups emerged, but simply used an a priori approach to create the four groups we had discussed in a previous, conceptual paper (Newall & Menec, 2018). We recognize that there are many other studies that could be conducted to further explore the four groups. We have added the idea of conducting cluster analysis in the Discussion section (p. 35). 

7. Are the variables marital status and living arrangements (e.g. living alone vs with others) available in this dataset? These variables are quite important for studies on social isolation and loneliness.

Yes, these variables are in the dataset and were used to create the social isolation variable. See p. 14.

Results

8. Could graphs be included for all outcome measures?

We have added graphs for desire for social participation and psychological distress in Supplemental Figure 1.

Discussion:

9. The take-aways, whether in terms of characterizations of or intervention for each group, needs to be made clearer. Currently, the findings read like lists of observations about each group. It will help the readers digest findings if the observations are organized in a more structured and coherent manner.

We have revised the Discussion and hope this addresses the Reviewer’s concerns.

10. Technology-based interventions for the vulnerable group does not sound realistic or sustainable, given that this group has the largest proportion of “low education” and “low income”. Will regular home visits be a more suitable intervention?

Technology-based interventions do not need to be expensive or difficult – and can be as simple as using the telephone. We have clarified this in the Discussion (p. 31). Home visits are an option, but would be more resource intensive. We have added home visits as another option in the Discussion (p. 31). 

11. Line 534 – 537: “Given that individuals in this group do not have their social needs met, despite having a relatively large social network, they may benefit from developing new friendships.” This sentence does not make sense. Why give people more of what is currently not working for them? It needs to be made clearer how the new friendships are different from existing ones.

We have revised this sentence to make it clearer that we mean developing new, meaningful social relationships (p. 31).

12. Line 537: It is curious that the line “As for the vulnerable group…” appeared in the middle of a paragraph ostensibly about the lonely in a crowd group. Is the content in the rest of the paragraph that appears after this line about the vulnerable group or lonely in a crowd group?

We have revised this sentence (p. 38).

13. What are the intervention implications of the results from age-based sensitivity analysis for older adults?

We have added a comment regarding this (p. 34). 

Reviewer #3: It is a great paper about the association of social isolation and loneliness with psychological distress, social support and civic participation. The topic is very interesting and there is a lack in the literature. However some aspects sohould improve:

1.-I think that the introduction should be shorter and focused on the topic (the association between social isolation and loneliness and their effect on civic participation, social support and health)

In response to the Reviewer’s comment, we have streamlined the introduction. 

2.-It would be more clear to call the groups: 1)Neither lonely nor isolated 2)Only isolated 3)Only lonely 4)lonely and isolated.

We have adopted the suggested terminology throughout the paper. 

3.-The authors should clarify the statistical analysis. I think that they used chi squared and anova tests for the comparisons in the descriptive analysis, adjusted and unadjusted ordered logistic regresssions models with the social support outcomes and, finally, adjusted and unadjusted logistic regressions models with participation and psychilogical distress as outcome. I also think that they used diferen sratifications of the samle to compare the effect between sepcific groups. But according to the methods section all this things are not clear.

We have revised the Analytic approach section to clarify the analyses.

4.-Authors should also clarify why the analysis wit psicological distress as outcome is prospective (like they say in the introduction)

The reason why psychological distress, but not desire for social participation and social support was analyzed prospectively was simply because of data availability. We have clarified this in the introduction (p. 4).

5.-In my opnion, the main results are that loneliness but not social isolation showed a significant effect on psychological destress as well as that the effect of loneliness seem to be greater than those of social isolation in most of the outcomes.

We agree with the Reviewer that loneliness has important health implication, beyond social isolation. We have added a comment to that effect in the Discussion (p. 31). 

6.-The authors should make further cites on rellevant literature such as:

We have added these citations.

Holwerda, T. J., Beekman, A. T. F., Deeg, D. J. H., Stek, M. L., van Tilburg, T. G., Visser, P. J., … Schoevers, R. A. (2012). Increased risk of mortality associated with social isolation in older men: only when feeling lonely? Results from the Amsterdam Study of the Elderly (AMSTEL). Psychological Medicine, 42(4), 843–853. https://doi.org/10.1017/S0033291711001772

Holwerda, T. J., Deeg, D. J. H., Beekman, A. T. F., van Tilburg, T. G., Stek, M. L., Jonker, C., & Schoevers, R. a. (2014). Feelings of loneliness, but not social isolation, predict dementia onset: results from the Amsterdam Study of the Elderly (AMSTEL). Journal of Neurology, Neurosurgery, and Psychiatry, 85(2), 135–142.

Reviewer #4: Review for PLoS One Journal

Social Isolation and Loneliness Groups: Associations with Social Support and Psychological Distress using Canadian Longitudinal Study of Aging (CLSA) Data

The purpose of this paper was to combine measures of social isolation and loneliness to create different groups and determine their association with social support and psychological distress. The four groups created were the majority group, the loner group, the lonely in a crowd group, and the vulnerable group (individuals in this group are dually socially isolated and lonely). In general, the majority group seemed to have the most positive and beneficial associations with social support and psychological distress.

I applaud the authors of this paper as it is innovative and will contribute to the social isolation and loneliness literature. The authors are correct in that many peer-review manuscripts have not examine social isolation and loneliness together, and this aspect of the manuscript alone greatly contributes to its innovation.

In my review of your manuscript, I have found some critiques which I believe will enhance the quality of the manuscript. The biggest issue in my opinion is regarding the methods. More detailed explanations are necessary for the type of analysis utilized, and also differentiating between the cross-sectional analyses and the prospective analyses.

The critiques of your manuscript organized by each section of your paper, from the abstract to the conclusion. I have included line numbers for where I find these issues.

ABSTRACT:

In the methods subsection of the abstract, it would be helpful to include the type of analyses done in the paper. This includes listing which analyses are prospective and which analyses are cross-sectional, and the type of regression modeling utilized (ordinary least squares, logistic, etc.)

We have added this information in the abstract.

INTRODUCTION:

Line 97: This study is actually over 30 years old now, so it should read over 30 years.

Thanks for catching this mistake. We have revised this accordingly.

Lines 123-125: The authors raise a very important point here, in that most studies have not examined the interactive effects of social isolation and loneliness on health. That being said, the authors should also consider including studies by Cornwell and Waite (2009) and Coyle and Dugan (2012). These studies have simultaneously examined social isolation and loneliness (or Cornwell and Waite’s notion, social disconnectedness and perceived isolation) on health. Please include these studies when covering the literature.

We have added these citations. 

Lines 162-167: Steptoe and colleagues (2013) also examined an interaction effect between social isolation and loneliness on mortality, and the interaction (I believe) was found to be non-significant.

We have added this article. The study did, indeed, not find a significant interaction.

Lines 260-263: There needs to be substantially more information on the amount of missing here. In particular, which measure had the most missing and what measure had the least missing? Were the missing enough to substantially affect the analysis? In general, if the missing are greater than 10%, this can represent a significant problem in the analysis.

Also, how were missing data handled? (I assume through listwise deletion?) If this is the case, then it should be stated.

We have added a comment re missing data (which was less than 3%) and also now note that we used list-wise deletion of missing data (p. 19).

Also, a final sample size should be stated for each of the analyses.

We have added the unweighted sample size in the tables 3-5.

METHODS

In lines 354-363, the authors discuss having a baseline mental health variable to control for the effects of mental health at baseline. I agree with the authors and think, overall, it would be better to include psychiatric distress at baseline and as an outcome. That being said, I believe it would be helpful to include (in supplementary analyses or aside from the main analyses) another set of analyses without the baseline mental health variable. It would be helpful to do this to determine the consistency of the results and to determine if any of the relationships change.

We have conducted a sensitivity analysis that did not include baseline depression in the model. The results remain the same (p. 24). 

In lines 380-385: There needs to be a stronger description of the analytic methods used in these analyses. If these are prospective analyses, then that should be stated. For example, if this is a lagged variable analysis or hierarchical modeling analysis, then this should be stated as well.

We have revised the methods section to clarify the analyses (p. 18).

RESULTS

No changes necessary.

DISCUSSION

Overall, the discussion is written very well.

From lines 537 to 543 is somewhat of a weird transition. A few lines above, it starts with the most vulnerable group, and then it transitions to the lonely in a crowd group. From 537-543, it then goes back to the most vulnerable group. I would suggest discussing one group at a time instead of going back and forth between groups to increase clarity.

We had revised this section (p. 37). 

6. PLOS authors have the option to publish the peer review history of their article (what does this mean?). If published, this will include your full peer review and any attached files.

Do you want your identity to be public for this peer review? For information about this choice, including consent withdrawal, please see our Privacy Policy.

Reviewer #1: No

Reviewer #2: No

Reviewer #3: No

Reviewer #4: No

---

## [Decision Letter · Decision Letter 1]

27 Feb 2020

PONE-D-19-26452R1

Examining social isolation and loneliness together in relation to social support and psychological distress using Canadian Longitudinal Study of Aging (CLSA) data

PLOS ONE

Dear Dr. Menec,

Thank you for submitting your manuscript to PLOS ONE. After careful consideration, we feel that it has merit but does not fully meet PLOS ONE’s publication criteria as it currently stands. Therefore, we invite you to submit a revised version of the manuscript that addresses the points raised during the review process.

Please proofread the manuscript very carefully and address the minor changes suggested by reviewer 1. Once these minor edits are corrected, the manuscript will be accepted for publication.

We would appreciate receiving your revised manuscript by Apr 12 2020 11:59PM. To enhance the reproducibility of your results, we recommend that if applicable you deposit your laboratory protocols in protocols.io, where a protocol can be assigned its own identifier (DOI) such that it can be cited independently in the future. For instructions see: http://journals.plos.org/plosone/s/submission-guidelines#loc-laboratory-protocols

We look forward to receiving your revised manuscript.

Kind regards,

Simone Reppermund, PhD

Academic Editor

PLOS ONE

Reviewers' comments:

Reviewer's Responses to Questions

**Comments to the Author**

1. If the authors have adequately addressed your comments raised in a previous round of review and you feel that this manuscript is now acceptable for publication, you may indicate that here to bypass the “Comments to the Author” section, enter your conflict of interest statement in the “Confidential to Editor” section, and submit your "Accept" recommendation.

Reviewer #1: All comments have been addressed

Reviewer #2: All comments have been addressed

Reviewer #4: All comments have been addressed

2. Is the manuscript technically sound, and do the data support the conclusions?

Reviewer #1: Yes

Reviewer #2: Yes

Reviewer #4: Yes

3. Has the statistical analysis been performed appropriately and rigorously? 

Reviewer #1: Yes

Reviewer #2: Yes

Reviewer #4: Yes

4. Have the authors made all data underlying the findings in their manuscript fully available?

Reviewer #1: (No Response)

Reviewer #2: Yes

Reviewer #4: Yes

5. Is the manuscript presented in an intelligible fashion and written in standard English?

Reviewer #1: Yes

Reviewer #2: Yes

Reviewer #4: Yes

6. Review Comments to the Author

Reviewer #1: Title: In my opinion, the word ‘together’ should be deleted.

Abstract

Line 20 – the word ‘respectively’ should be deleted.

Line 81 – Note that there are multiple references showing them to be strongly related to each other. I suggest either deleting this sentence or adding some references to the contrary as well.

Line 145. I recommend deleting the word “indeed”

Line 147-149 There are multiple studies showing a significant relationship between social isolation and loneliness. Those should also be cited.

Line 553 – you still use the ‘lonely in a crowd’ term, which should be changed.

Line 562 - “support with every tasks” – a word seems to be missing

Line 615. The first conclusion is simply self-congratulatory and could be deleted. The other conclusions should convince the reader whether this analysis was worth while, without the author telling the reader that.

In general – I was not able to read all sections very carefully, yet I found some needs for editorial changes. I suggest the author consider having an editor review the manuscript carefully to detect any other editorial needs.

Reviewer #2: (No Response)

Reviewer #4: (No Response)

7. PLOS authors have the option to publish the peer review history of their article (what does this mean?). If published, this will include your full peer review and any attached files.

Reviewer #1: No

Reviewer #2: No

Reviewer #4: No

---

## [Author Response · Author response to Decision Letter 1]

4 Mar 2020

PONE-D-19-26452R1

Examining social isolation and loneliness together in relation to social support and psychological distress using Canadian Longitudinal Study of Aging (CLSA) data

PLOS ONE

Dear Dr. Menec,

Thank you for submitting your manuscript to PLOS ONE. After careful consideration, we feel that it has merit but does not fully meet PLOS ONE’s publication criteria as it currently stands. Therefore, we invite you to submit a revised version of the manuscript that addresses the points raised during the review process.

Please proofread the manuscript very carefully and address the minor changes suggested by reviewer 1. Once these minor edits are corrected, the manuscript will be accepted for publication.

Dear Dr. Reppermund;

We have made the suggested edits and have also reviewed the paper for typos. See below for responses to the suggested edits.

Sincerely,

Verena Menec, PhD

RESPONSE: ==============================

We would appreciate receiving your revised manuscript by Apr 12 2020 11:59PM. To enhance the reproducibility of your results, we recommend that if applicable you deposit your laboratory protocols in protocols.io, where a protocol can be assigned its own identifier (DOI) such that it can be cited independently in the future. For instructions see: http://journals.plos.org/plosone/s/submission-guidelines#loc-laboratory-protocols

• A rebuttal letter that responds to each point raised by the academic editor and reviewer(s). This letter should be uploaded as separate file and labeled 'Response to Reviewers'.

• A marked-up copy of your manuscript that highlights changes made to the original version. This file should be uploaded as separate file and labeled 'Revised Manuscript with Track Changes'.

• An unmarked version of your revised paper without tracked changes. This file should be uploaded as separate file and labeled 'Manuscript'.

We look forward to receiving your revised manuscript.

Kind regards,

Simone Reppermund, PhD

Academic Editor

PLOS ONE

Reviewers' comments:

Reviewer's Responses to Questions

Comments to the Author

1. If the authors have adequately addressed your comments raised in a previous round of review and you feel that this manuscript is now acceptable for publication, you may indicate that here to bypass the “Comments to the Author” section, enter your conflict of interest statement in the “Confidential to Editor” section, and submit your "Accept" recommendation.

Reviewer #1: All comments have been addressed

Reviewer #2: All comments have been addressed

Reviewer #4: All comments have been addressed

2. Is the manuscript technically sound, and do the data support the conclusions?

Reviewer #1: Yes

Reviewer #2: Yes

Reviewer #4: Yes

3. Has the statistical analysis been performed appropriately and rigorously? 

Reviewer #1: Yes

Reviewer #2: Yes

Reviewer #4: Yes

4. Have the authors made all data underlying the findings in their manuscript fully available?

Reviewer #1: (No Response)

Reviewer #2: Yes

Reviewer #4: Yes

5. Is the manuscript presented in an intelligible fashion and written in standard English?

Reviewer #1: Yes

Reviewer #2: Yes

Reviewer #4: Yes

6. Review Comments to the Author

Reviewer #1: Title: In my opinion, the word ‘together’ should be deleted.

We have changed the wording to “in combination”

Abstract

Line 20 – the word ‘respectively’ should be deleted.

This change has been made.

Line 81 – Note that there are multiple references showing them to be strongly related to each other. I suggest either deleting this sentence or adding some references to the contrary as well.

We have edited the sentence to say that several studies show that social isolation and loneliness are only weakly correlated. 

Line 145. I recommend deleting the word “indeed”

This change has been made.

Line 147-149 There are multiple studies showing a significant relationship between social isolation and loneliness. Those should also be cited.

The point in this section was not to discuss how social isolation and loneliness are related to each other, but rather how the interaction between the two variables is related to mortality. We have edited the sentence to make this clearer. 

Line 553 – you still use the ‘lonely in a crowd’ term, which should be changed.

This change has been made.

Line 562 - “support with every tasks” – a word seems to be missing

This change has been made.

Line 615. The first conclusion is simply self-congratulatory and could be deleted. The other conclusions should convince the reader whether this analysis was worth while, without the author telling the reader that.

We have revised this sentence.

In general – I was not able to read all sections very carefully, yet I found some needs for editorial changes. I suggest the author consider having an editor review the manuscript carefully to detect any other editorial needs.

We have reviewed the paper for typos.

Reviewer #2: (No Response)

Reviewer #4: (No Response)

---

## [Editor Report · Decision Letter 2]

6 Mar 2020

Examining social isolation and loneliness in combination in relation to social support and psychological distress using Canadian Longitudinal Study of Aging (CLSA) data

PONE-D-19-26452R2

Dear Dr. Menec,

We are pleased to inform you that your manuscript has been judged scientifically suitable for publication and will be formally accepted for publication once it complies with all outstanding technical requirements.

With kind regards,

Simone Reppermund, PhD

Academic Editor

PLOS ONE
---

## [Editor Report · Acceptance letter]

10 Mar 2020

PONE-D-19-26452R2 

Examining social isolation and loneliness in combination in relation to social support and psychological distress using Canadian Longitudinal Study of Aging (CLSA) data 

Dear Dr. Menec:

I am pleased to inform you that your manuscript has been deemed suitable for publication in PLOS ONE. Congratulations! Your manuscript is now with our production department. 

With kind regards,

on behalf of

Dr. Simone Reppermund 

Academic Editor

PLOS ONE